# SINGER: Leveraging Semantic Identifier Hierarchies for Generative Recommendation

## Abstract

Recent advances in large language models (LLMs) have sparked a new line of generative recommendation, where the recommender autoregressively generates a sequence of Semantic IDs (SIDs)—item identifiers in the SID space—rather than ranking a pre-selected candidate set of item titles in the language space. Although the current Supervised Fine-Tuning followed by Reinforcement Learning (SFT-then-RL) pipeline improves performance, it still fails to adequately model the SID space. Specifically, (i) SFT often leads to superficial SID understanding by merely forcing memorization of a closed SID vocabulary, and (ii) rule-based RL typically relies on coarse-grained rewards that treat all incorrect SIDs equally, regardless of their hardness. To address these challenges, we propose *SID-Navigated GEnerative Recommender* (**SINGER**), a framework that integrates fine-grained SID knowledge throughout training. SINGER comprises two components: (1) Full-Process SID Alignment, which embeds alignment objectives throughout both SFT and RL to strengthen the model's understanding of the SID space; (2) SID-Navigated Reinforcement Learning, which consists of SID-level rewards that grade each trajectory by the deepest correctly matched SID layer, together with a SID-prefix curriculum sampling strategy that supplies partial prefixes as intermediate guidance for hard cases. Experiments on public benchmarks demonstrate that SINGER consistently outperforms strong sequential, generative, and recent LLM-based baselines across standard metrics, validating the benefit of integrating hierarchical SID signals with the world knowledge of pretrained LLMs.

## 1 Introduction

The powerful sequence modeling capabilities of large language models (LLMs) have enabled their adaptation to recommender systems (Bao et al., 2023; Sheng et al., 2024; Hu et al., 2025; He et al., 2025a; Wu et al., 2024; Fang et al., 2020), with generative recommendation emerging as a promising direction that leverages autoregressive generation for item prediction (Rajput et al., 2023; Zheng et al., 2024; Qu et al., 2024; Zhai et al., 2024; Deng et al., 2025; Wang et al., 2025a). This paradigm centers on generating sequences of Semantic IDs (SIDs)—discrete tokens that encode item semantics through quantization of continuous embeddings (Zeghidour et al., 2022; Luo et al., 2024). By promoting token sharing across semantically related items, SIDs facilitate efficient handling of large-scale catalogs while naturally aligning with the step-by-step (chain-of-thought) reasoning paradigm of LLMs (Rajput et al., 2023; Zeghidour et al., 2022; Luo et al., 2024; Deng et al., 2025; Singh et al., 2024).

Upon scrutinizing prior studies on generative recommenders, we can summarize a common training pipeline: (1) Beginning with items' textual descriptions or embeddings, a quantization method transforms continuous vectors into SIDs. (2) The model is then trained to generate these SIDs in an end-to-end manner, typically following two primary approaches: *scratch-trained recommenders* that train Transformer (Vaswani et al., 2017) from scratch on user interaction sequences (Rajput et al., 2023; Zhai et al., 2024; Deng et al., 2025; Wang et al., 2025a), and *SID-aligned Recommenders* that adapt pretrained LLMs from the language space into SID space through supervised fine-tuning (SFT) (Zheng et al., 2024; Qu et al., 2024). Although these methods achieve promising performance, the integration of reinforcement learning (RL) for deeper alignment with user interaction sequences remains relatively underexplored (Chen et al., 2024).

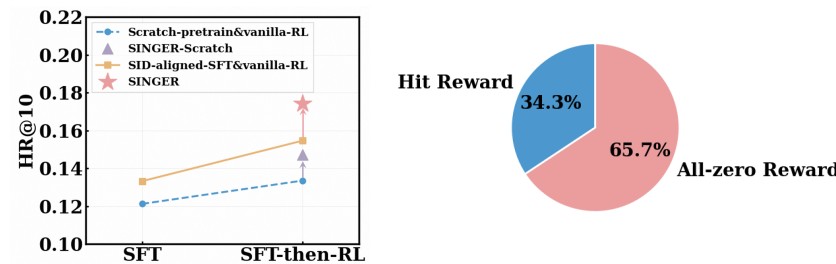

(a) Performance under Different Train-
ing Paradigms.

(b) Sample hit reward distribution.

Figure 1: Results on the Industrial dataset. Figure 1a illustrates the performance, with respect to
HitRatio@10 under different training paradigms. Figure 1b shows the proportion of RL-sampled
outputs from the SFT-initialized model that receive a non-zero reward.

Inspired by the recent success of the **SFT-then-RL** paradigm (Shao et al., 2024; Yoshihara et al.,
2025; OpenAI et al., 2024), we aim to establish a generative recommendation framework that adapts
this paradigm to the unique characteristics of recommendation tasks. Specifically, we first explore
applying Group Relative Policy Optimization (GRPO) (DeepSeek-AI et al., 2025) after the initial
SFT phase, considering both scratch-trained and SID-aligned recommenders. Our preliminary
experiments (*cf.* Figure 1a) demonstrate the effectiveness of the SFT-then-GRPO paradigm in
generative recommendation, consistently outperforming the SFT-only baselines. Building on this,
we further investigate two critical limitations that arise when directly transferring the standard
SFT-then-RL paradigm to recommendation, which constrain its full potential in this domain:

• **Limited SID Understanding in Alignment**. The SFT alignment process constrains LLMs to
  project outputs into a closed SID vocabulary through supervised training on user-item interaction
  sequences. Such rigid constraint mainly encourages superficial pattern matching rather than true
  semantic understanding of SIDs. As illustrated in Figure 2, even after SFT the model often fails to
  exploit SID histories correctly: it either resorts to generic responses citing insufficient user infor-
  mation, or produces lengthy, repetitive item descriptions indicative of collapsed generation. While
  the world knowledge in pretrained LLMs should be beneficial for understanding item semantics
  and user preference (*cf.* Figure 1a), the current alignment pipeline exploits it only superficially,
  highlighting the need for a more systematic approach that integrates SID understanding throughout
  the entire training process.

• **Ineffective Reward Assignment in RL**. The standard rule-based RL training treats each SID
  sequence as an indivisible unit—if any token in a generated sequence fails to match the ground-
  truth, the entire sequence is penalized with zero reward. This binary reward mechanism overlooks
  the rich relational structure among SIDs, unable to distinguish between near-correct and completely
  irrelevant predictions. The resulting sparse rewards (*cf.* Figure 1b) deprives the model of useful
  learning signals on hard cases—those rollouts that narrowly miss exact matches and thus receive
  zero advantages (Yu et al., 2025)—which represent the most critical cases for developing genuine
  reasoning capabilities over the SID taxonomy. Consequently, RL optimization tends to reinforce
  patterns already memorized during pre-training and SFT (Chu et al., 2025; Yue et al., 2025; Liu
  et al., 2025), rather than learning to navigate semantic relationships that would enable better
  performance on challenging cases.

To address the above limitations, we introduce *SID-Navigated GEnerative Recommenders* (**SINGER**),
a generative recommendation framework that enhances both SID comprehension and reward utiliza-
tion throughout the SFT-then-RL process. SINGER is built on two key components:

• **Full-Process SID Alignment.** We embed a set of alignment objectives throughout the entire
  SFT-then-RL pipeline to achieve deeper SID alignment. Specifically, we integrate discrete SID
  tokens into the LLM's vocabulary and introduce a series of auxiliary alignment tasks (*e.g.,* explicit
  item title to SID mapping) that are enforced during both SFT and RL phases. This ensures the
  model fully internalizes the structural semantics of SIDs.

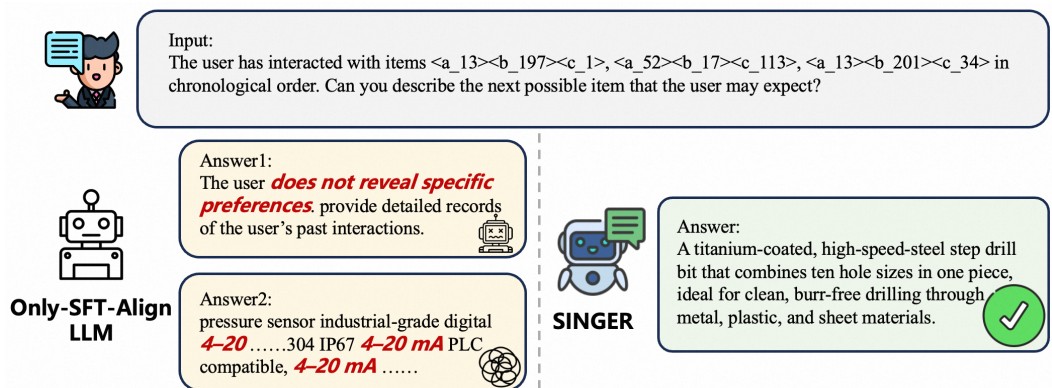

Figure 2: Only-SFT-Align case study. The model is fed a SID-formatted interaction history and asked to describe the next item. The Only-SFT-Align LLM fails to interpret the SID tokens (Answer 1) or generates verbose, repetitive, and disordered text (Answer 2), underscoring the shallow alignment achieved by SFT alone. By contrast, SINGER correctly understands the SID sequence and produces a concise, coherent item description.

- **SID-Navigated Reinforcement Learning.** We develop a novel RL framework that leverages SID hierarchy to provide more informative learning signals:

  (a) **SID-Prefix Curriculum Sampling.** Let the ground-truth item be a hierarchical token sequence $e^{pos} = (s_a^\star, s_b^\star, s_c^\star)$. Inspired by curriculum learning, we define a scheduling function $f(t, e^{pos})$ that truncates $e^{pos}$ and progressively shortens the retained prefix as the training step $t$ increases. The resulting truncated prefix is concatenated with the original input to construct a new prompt. The policy then produces a continuation and the reward is computed accordingly. In this way, the model starts by predicting lower-level tokens conditioned on higher-level prefixes and gradually transitions to prefix-free generation, thereby achieving autonomous exploration and performance gains on difficult samples.

  (b) **SID-Level Reward Modeling.** We employ GRPO to optimize the LLM with fine-grained, SID-level rewards. For hard samples whose original rewards are often zero, we treat the step-by-step generation of the SID path $\langle a \rangle \to \langle b \rangle \to \langle c \rangle$ as an intermediate reasoning process. Suppose the ground-truth SID token be $e^{pos} = (s_a^\star, s_b^\star, s_c^\star)$ and the model output be $e = (s_a, s_b, s_c)$. A partial reward is granted according to the deepest level that still matches the ground truth, *i.e.*, $\kappa(e, e^{pos}) = \max\{ k : e_{a:k} = e_{a:k}^\star \}$, preventing zero-reward collapse on hard examples.

We evaluate SINGER on two public benchmark datasets (Hou et al., 2024) and compare it with a wide range of leading traditional sequential recommenders, generative recommenders, and several recent LLM-based baselines. The results show that SINGER consistently outperforms all competitors on standard recommendation metrics, confirming its effectiveness. *A detailed survey of related work is deferred to Appendix A.*

## 2 PRELIMINARY

### 2.1 RQ-KMEANS

For each item $i$, we concatenate its title and textual description and feed the resulting sentence into a frozen content encoder to produce a $d$-dimensional semantic vector $\mathbf{x} \in \mathbb{R}^d$. The continuous vector is then discretized with the RQ-Kmeans algorithm (Luo et al., 2024), which builds a hierarchy of codebooks by recursively clustering the residuals.

Let $\tilde{\mathbf{M}} = [\mathbf{x}_1; \ldots; \mathbf{x}_N] \in \mathbb{R}^{N \times d}$ be the matrix that stacks the embeddings of all $N$ items. We initialize $\mathbf{R}^{(1)} = \tilde{\mathbf{M}}$. For each layer $l \in \{1, \ldots, L\}$, where $L$ is the number of hierarchical levels in the semantic codebook, we learn a codebook $\mathbf{C}^{(l)} = \{\mathbf{c}_k^{(l)}\}_{k=1}^{K_l}$ by running K-means with $K_l$ centroids on the current residuals $\mathbf{R}^{(l)}$:

$$\mathbf{C}^{(l)} = \text{K-means}(\mathbf{R}^{(l)}, K_l).$$

For item $i$ ($1 \leq i \leq N$), the index of the nearest centroid is obtained via

$$s_i^{(l)} = \arg\min_k \|\mathbf{R}_i^{(l)} - \mathbf{c}_k^{(l)}\|_2,$$

where $\|\cdot\|$ denotes the Euclidean norm. The residual is updated as

$$\mathbf{R}_i^{(l+1)} = \mathbf{R}_i^{(l)} - \mathbf{c}_{s_i^{(l)}}^{(l)}.$$

After $L = 3$ layers we obtain a coarse-to-fine set of semantic identifiers, $\{s_i^{(1)}, s_i^{(2)}, s_i^{(3)}\}$, which serves as the unique token sequence for item $i$ and will be consumed by the recommender for progressive generation.

## 2.2 GROUP-RELATIVE POLICY OPTIMIZATION

We fine-tune our policy with the GRPO algorithm (Shao et al., 2024; DeepSeek-AI et al., 2024), which leverages the relative quality of multiple responses generated for the same prompt. Concretely, for each input $x \sim D$, we roll out the current policy $\pi_{\theta_{old}}$ $G$ times to obtain a set of candidates $\mathcal{Y}(x) = \{y^{(1)}, \ldots, y^{(G)}\}$. Each candidate $y^{(i)}$ receives a scalar reward $R_i$, and the advantage is computed by normalizing the rewards *within the group*

$$\hat{A}_i = \frac{R_i - \text{mean}(R_{1:G})}{\text{std}(R_{1:G})}, \tag{1}$$

where $R_{1:G} = \{R_1, \ldots, R_G\}$. This groupwise normalization recenters advantages at zero and rescales them to unit variance, thereby turning each prompt into a self-contained comparison game and reducing gradient variance. The new policy $\pi_\theta$ maximizes the clipped surrogate

$$J_{\text{GRPO}}(\theta) = \mathbb{E}_{x \sim D, \, y^{(i)} \sim \pi_{\theta_{old}}} \left[ \frac{1}{G} \sum_{i=1}^{G} \frac{1}{|y^{(i)}|} \sum_{t=1}^{|y^{(i)}|} \left\{ \min\left(r_{i,t}\hat{A}_{i,t}, \, \text{clip}(r_{i,t}, 1-\epsilon, 1+\epsilon)\hat{A}_{i,t}\right) \right. \right.$$
$$\left. \left. - \beta \, \text{KL}\left[\pi_\theta \| \pi_{\text{ref}}\right] \right\} \right], \tag{2}$$

where $r_{i,t} = \frac{\pi_\theta(y_t^{(i)} \mid x, y_{<t}^{(i)})}{\pi_{\theta_{old}}(y_t^{(i)} \mid x, y_{<t}^{(i)})}$ is the per-token importance ratio, $\epsilon$ is the clipping threshold. The term $\beta$ balances the task reward against a KL penalty, keeping the updated policy close to the reference model $\pi_{\text{ref}}$, which is the frozen initial SFT policy.

## 2.3 TASK FORMULATION

Generative recommendation reformulates the recommendation problem as a sequence generation task. Let $H_u$ denote the interaction history of user $u$, sorted in chronological order. Each item $i \in H_u$ is represented by a 3–level SID tuple $\{s_i^{(1)}, s_i^{(2)}, s_i^{(3)}\}$. Given $H_u$, the generative recommender $\pi_\theta$, parameterized by $\theta$, is trained to predict an item $i^{\text{pos}}$ that best matches the preferences of user $u$ from the item set. During inference, we employ beam search to generate the top-$k$ candidates and evaluate the model with standard generative-recommendation metrics.

## 3 METHODOLOGY

To address the key shortcomings of the existing SFT-then-RL pipeline for generative recommendation, we propose **SINGER**. The framework elevates the performance ceiling by aligning the LLM with the SID space throughout the entire training process and applying SID-navigated optimization in the RL stage, as illustrated in Figure 3.

## 3.1 FULL-PROCESS SID ALIGNMENT

As demonstrated in Section 1, aligning world knowledge with item SIDs is beneficial for generative recommendation (Zheng et al., 2024). Hence, instead of the paradigm that trains only on SIDs

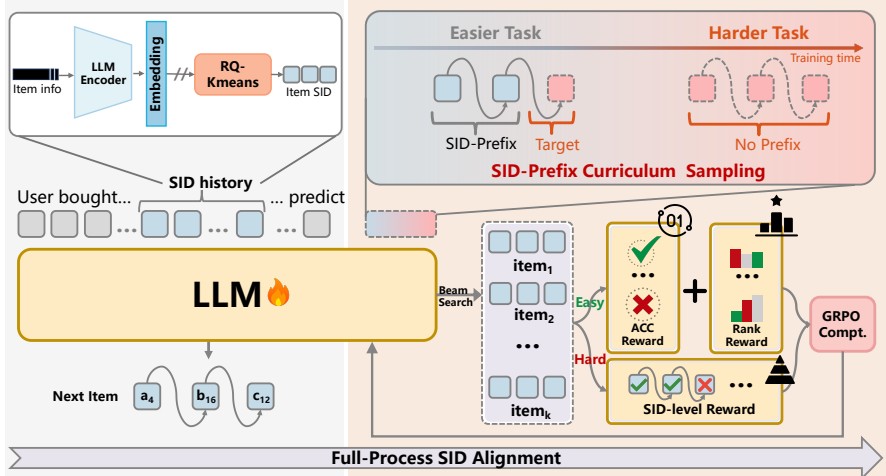

Figure 3: SINGER framework. RQ-Kmeans builds the item SID codebook and SFT first aligns the LLM. In RL, beam search with a SID-prefix curriculum progressively shortens the given prefix, thereby hardening the task while matching the inference setup. Hit cases receive an accuracy–rank reward, whereas a SID-level reward grants partial credit to semantically close SIDs when no hit is found, alleviating sparse feedback. GRPO updates the policy, and SID alignment is enforced end-to-end.

(Rajput et al., 2023; Deng et al., 2025; Wang et al., 2025a), our LLM-based recommender explicitly strengthens the link between language understanding and collaborative semantics by injecting a set of alignment objectives:

- **Semantic Tasks.** Given a chronologically ordered sequence of historical SIDs and an explicit task instruction, the LLM is asked to predict the SID of the next item the user is likely to interact with.
- **Alignment Tasks.** It comprises a series of alignment tasks between the textual space and the SID space. Through these tasks, we encourage a bidirectional mapping that grounds SIDs in language and injects linguistic knowledge into SID representations.

Representative tasks from each category are jointly optimized throughout the entire SFT-then-RL pipeline, enabling the LLM to fully exploit its world knowledge, deepen its understanding of SIDs, and ultimately boost generative-recommendation performance. During the RL phase, we employ constrained decoding, limiting the output space to a precompiled dictionary that contains the SID of each item as well as its canonical title. This restriction ensures that the LLM can emit only legal identifiers, making it straightforward to compute a rule-based, verifiable reward signal. Detailed examples of the prompts are provided in the Appendix D.

## 3.2 SID-NATIGATED REINFORCEMENT LEARNING

To fully exploit the fine-grained signals carried by SIDs, we introduce SID-Natigated Reinforcement Learning (SIN-RL). SIN-RL comprises two complementary components—curriculum sampling and reward modeling. By leveraging the codebook's inherent coarse-to-fine hierarchy, SIN-RL steers the agent toward harder examples in a structured manner, thereby improving its capability in the challenging regions of the data distribution.

### 3.2.1 SID-PREFIX CURRICULUM SAMPLING

We first optimize the sampling strategy to preserve rollout diversity in RL for generative recommendation. In conventional LLM–RL training, LLM first applies dynamic sampling to obtain several candidate outputs and then computes a group–level reward. Recent studies, however, report a rapid entropy drop during RL, which harms diversity (Wang et al., 2025b; Cui et al., 2025; He et al., 2025b; Mukherjee et al., 2025). Inspired by these findings, we use a new sampling strategy tailored to generative recommendation so as to keep rollouts diverse. Specifically, when $k$ trajectories are required, we do not run $k$ independent dynamic samplings, each predicting a single item. Instead,

we execute one beam–search pass and take the top–$k$ item SIDs as the trajectory set for subsequent reward estimation. Following prior work (Bao et al., 2024), we remove length normalization in beam search to avoid bias amplification in the LLM-based recommender.

Furthermore, we label a training instance $(x, y^*)$ as difficult when none of the $k$ items generated in the current rollout matches the ground-truth SIDs $y^*$. Once normal and hard tasks are separated, every hard sample $(x, y^*) \in \mathcal{D}_{\text{sin}}$ is handled with a SID-Prefix curriculum schedule. At the beginning of training, we expose a long SID prefix; as learning progresses, the length of this prefix is gradually shortened, encouraging the model to explore on its own. Formally, we use p(t) to control the length of the SID-prefix:

$$p(t) = 1 - \frac{t}{T} \tag{3}$$

where $T$ is the total number of RL steps and $t$ is the current step. Therefore, the value p(t) decreases linearly from 1 to 0, so the depth of the prefix guidance diminishes accordingly throughout training.

For every hard instance $(x, y^*) \in \mathcal{D}_{\text{sin}}$, we first compute the length of the SID-prefix, $L_{\text{guide}}$, according to the schedule $p(t)$:

$$L_{guide} = \lfloor p(t) \cdot L \rfloor \tag{4}$$

where $\lfloor \cdot \rfloor$ is the floor operator. We then truncate $y^*$ to its first $L_{\text{guide}}$ tokens, denoted by $y^*_{L_{\text{guide}}}$, and concatenate it with the original input $x$ to form a new prompt:

$$x_{\text{sin}} = x \oplus y^*_{L_{guide}} \tag{5}$$

Finally, the policy produces a continuation $y \sim \pi_\theta(\cdot \mid x_{\text{sin}})$, upon which the RL reward is evaluated.

### 3.2.2 SID-LEVEL REWARD MODELING

For a sampled item $e_i$ generated by the model, the naive rule-based reward $R_{\text{acc}}$ follows a binary scheme. The ground-truth item $e^{\text{pos}}$ is assigned 1, whereas every other candidate receives 0, as shown below:

$$R_{\text{acc}}(e_i) = \begin{cases} 1, & e_i = e^{\text{pos}}, \\ 0, & \text{otherwise.} \end{cases} \tag{6}$$

Such sparsity treats all negative samples equally and thus fails to reflect their different levels of hardness. We first follow the recent paradigm by constructing an auxiliary ranking score $R_{\text{rank}}$ that exploits the ordering information among candidate items. Specifically, a negative sample that appears higher in the generation list (i.e. is produced with a larger probability) should be penalized more. Let $p_i$ denote the position of a negative item $e_i^{\text{neg}}$. Its ranking reward is defined as the negative reciprocal of the natural logarithm of $(p_i + 1)$, while the correct item $e^{\text{pos}}$ is given 0:

$$R_{\text{rank}}(e_i) = \begin{cases} 0, & e_i = e^{\text{pos}}, \\ -\dfrac{1}{\log(p_i + 1)}, & \text{otherwise.} \end{cases} \tag{7}$$

Although the ranking reward supplies denser guidance, it still evaluates the whole sequence $\langle a \rangle \langle b \rangle \langle c \rangle$ as a single target and overlooks the multi-granular clues embedded in the SID codebook. Consequently, for many difficult instances $(x, y^*) \in \mathcal{D}_{\text{sin}}$ the reward may still collapse to zero, leaving the sparsity issue unsolved.

To address this limitation, we further view the generation of the hierarchical SID $\langle a \rangle \to \langle b \rangle \to \langle c \rangle$ as a step-by-step inference process and design SID-level reward. Instead of relying on external Process-Reward Models, the proposed approach harnesses the codebook's innate coarse-to-fine hierarchy to provide intermediate reasoning signals at each SID layer. This objective can be formally expressed as:

$$R_{\text{reason}}(e_i) = 1 - \lambda^k \tag{8}$$

where $k = f(e_i, e^{pos})$ denotes the deepest layer at which the generated sequence $e_i$ matches the ground-truth sequence $e^{pos}$, and $\lambda \in (0, 1)$ is a decay coefficient that modulates the reward increment across different layers and guarantees that the overall reward always lies in the interval $0 < R_{reason}(e_i) < 1$. The final total reward can be formally expressed as:

$$R(e_i) = \begin{cases} R_{\text{acc}}(e^i) + R_{\text{rank}}(e^i), & (x, y^*) \notin \mathcal{D}_{\text{sin}}, \\ R_{\text{reason}}(e^i), & \text{otherwise.} \end{cases} \tag{9}$$

For challenging samples $(x, y^*) \in \mathcal{D}_{\text{sin}}$, SID-Level Reward Modeling fully exploits SID's hierarchical rewards to deliver fine-grained guidance.

## 4 EXPERIMENTS

In this section, we first report the empirical performance of **SINGER** on two real-world benchmarks (Hou et al., 2024), and compare it against a selected set of baselines covering conventional sequential recommenders, SID-based generative models, and recent LLM-powered recommenders. In addition, we recast the SID-based next-item prediction task as recommendation rule discovery and show that SINGER offers extra gains for cold-start users with scarce interactions and even for completely unseen domains. We further conduct extensive ablation studies to pinpoint the components that most contribute to SINGER's effectiveness. Refer to the appendix B for detailed implementation details. In short, this section is organized to answer the following research questions:

- **RQ1:** How does SINGER perform in comparison to other baseline methods?
- **RQ2:** How does SINGER perform under completely unseen domains?
- **RQ3:** How do the designed components contribute to SINGER's recommendation efficacy?

**Datasets and Metrics.** We conduct extensive experiments on two real–world subsets of the Amazon Review corpus—*Office* and *Industrial*. Following common practice, we adopt Hit Rate (HR@K) and Normalized Discounted Cumulative Gain (NDCG@K) to evaluate the top–$K$ recommendation accuracy. Please refer to Appendix B for more details about datasets and evaluation metrics.

**Baselines.** Our baselines contain three categories: (1) Traditional recommendation models, including GRU4Rec (Hidasi et al., 2016), Caser (Tang and Wang, 2018), SASRec (Kang and McAuley, 2018); (2) Generative recommendation models: HSTU (Zhai et al., 2024), TIGER (Rajput et al., 2023), LC-Rec (Zheng et al., 2024); (3) LLM-based recommendation models, including BigRec (Bao et al., 2023), D3 (Bao et al., 2024), S-DPO (Chen et al., 2024). Please see Appendix B for more information.

### 4.1 PERFORMANCE COMPARISON (RQ1)

We conduct a comprehensive evaluation of SINGER on three benchmark datasets—Industrial and Toys; the results are summarized in Table 1. Two major observations emerge:

- **Utility of LLM World Knowledge.** LLM-based recommenders such as BIGRec and D$^3$ markedly outperform classical paradigms like GRU4Rec and Caser, confirming that injecting the broad world knowledge encoded in LLMs can substantially boost recommendation quality.
- **Effectiveness of SINGER.** By incorporating fine-grained SID information into the RL loop and aligning the entire generation trajectory with the task objective, SINGER establishes new SOTA across all three datasets, significantly surpassing the strongest prior baselines.

### 4.2 UNSEEN-DOMAIN PERFORMANCE EVALUATION (RQ2)

To assess SINGER's generalization to out-of-distribution (OOD) data, we conduct an unseen-domain study termed *SID pattern discovery*. Specifically, the model is trained on the source domain Industrial and evaluated on a completely unseen target domain Office. Given that prior studies have shown SFT can overfit to the training domain and degrade OOD performance (Jin et al., 2025; Yue et al., 2025; Yoshihara et al., 2025; Cheng et al., 2025), we introduce an RL-only variant, SINGER-w/ RL, specifically to provide a version focused on OOD generalization. We benchmark three systems: (1) GRU4Rec, trained and tested on Office. (2) Qwen-Text, which encodes the user's interaction history as plain text and predicts the next item in textual form; (3) Qwen-SID, which represents the same history as a sequence of SIDs and predicts the next SID; (4) SINGER-w/ RL, which is trained solely with RL on Industrial and is tested on Office without SFT.

Table 1: Performance of SINGER Compared to Traditional Methods, Generative Methods, and LLM-based Methods

| Dateset | Methods | HR@3 | NDCG@3 | HR@5 | NDCG5 | HR@10 | NDCG@10 |
|---|---|---|---|---|---|---|---|
| | **Traditional** | | | | | | |
| | GRU4Rec | 0.0638 | 0.0542 | 0.0774 | 0.0598 | 0.0999 | 0.0669 |
| | Caser | 0.0618 | 0.0514 | 0.0717 | 0.0555 | 0.0942 | 0.0628 |
| | SASRec | 0.0790 | 0.0700 | 0.0909 | 0.0748 | 0.1088 | 0.0806 |
| | **Generative** | | | | | | |
| | HSTU | 0.0927 | 0.0885 | 0.1037 | 0.0918 | 0.1163 | 0.0958 |
| Industrial | TIGER | 0.0852 | 0.0742 | 0.1010 | 0.0807 | 0.1321 | 0.0908 |
| | LCRec | 0.0915 | 0.0805 | 0.1057 | 0.0862 | 0.1332 | 0.0952 |
| | **LLM-based** | | | | | | |
| | BIGRec | 0.0931 | 0.0841 | 0.1092 | 0.0907 | 0.1370 | 0.0997 |
| | $D^3$ | 0.1024 | 0.0991 | 0.1213 | 0.0989 | 0.1500 | 0.1082 |
| | S-DPO | 0.1032 | 0.0906 | 0.1238 | 0.0991 | 0.1524 | 0.1082 |
| | **Ours** | | | | | | |
| | SINGER | **0.1256** | **0.1112** | **0.1453** | **0.1192** | **0.1744** | **0.1276** |
| | **Traditional** | | | | | | |
| | GRU4Rec | 0.0629 | 0.0528 | 0.0789 | 0.0595 | 0.1019 | 0.0669 |
| | Caser | 0.0748 | 0.0615 | 0.0865 | 0.0664 | 0.1093 | 0.0737 |
| | SASRec | 0.0861 | 0.0769 | 0.0949 | 0.0805 | 0.1120 | 0.0858 |
| | **Generative** | | | | | | |
| | HSTU | 0.1134 | 0.1031 | 0.1252 | 0.1079 | 0.1400 | 0.1126 |
| Office | TIGER | 0.0986 | 0.0852 | 0.1163 | 0.0960 | 0.1408 | 0.1002 |
| | LCRec | 0.0921 | 0.0807 | 0.1048 | 0.0859 | 0.1237 | 0.0920 |
| | **LLM-based** | | | | | | |
| | BIGRec | 0.1069 | 0.0961 | 0.1204 | 0.1017 | 0.1434 | 0.1091 |
| | $D^3$ | 0.1204 | 0.1055 | 0.1406 | 0.1139 | 0.1634 | 0.1213 |
| | S-DPO | 0.1169 | 0.1033 | 0.1356 | 0.1110 | 0.1587 | 0.1255 |
| | **Ours** | | | | | | |
| | SINGER | **0.1331** | **0.1163** | **0.1472** | **0.1221** | **0.1746** | **0.1309** |

Table 2: Performance of SINGER and its variants on completely unseen recommendation domains

| Dataset | Method | HR@3 | NDCG@3 | HR@5 | NDCG@5 | HR@10 | NDCG@10 |
|---|---|---|---|---|---|---|---|
| | GRU4Rec | 0.0629 | 0.0528 | 0.0789 | 0.0595 | 0.1019 | 0.0669 |
| | Qwen-Text | 0.0031 | 0.0021 | 0.0044 | 0.0026 | 0.0057 | 0.0030 |
| Office | Qwen-SID | 0.0300 | 0.0214 | 0.0456 | 0.0282 | 0.0733 | 0.0373 |
| | SINGER-w/ RL | 0.0553 | 0.0433 | 0.0691 | 0.0489 | 0.0892 | 0.0553 |

As reported in Table 2, Qwen-Text performs rather poorly, whereas Qwen-SID is markedly better, demonstrating that the structured SID space is easier for a language model to exploit. Although SINGER-w/ RL lags behind the full SINGER on in-domain metrics, its RL-only optimization offers strong transferability, yielding competitive accuracy on the unseen Office catalogue surprisingly. Despite the substantial domain shift and the possible semantic mismatch among SIDs, SINGER can still discover transferable interaction patterns and produce high-quality recommendations for a brand-new catalogue, highlighting the encouraging unseen-domain potential of our framework.

## 4.3 ABLATION STUDY (RQ3)

To validate the effectiveness of each component in the SINGER framework, we compare it with the following alternative approaches.

### 4.3.1 ALIGNING STRATEGY

We benchmark the full model against three carefully designed variants: (1) SINGER−W/O ALIGN: A pure *SID→SID* paradigm: the input consists of SID-organized user histories, and the target is the SID of the next item. No cross-modal alignment is applied in either stage. (2) SINGER−W/ SFTALIGN:

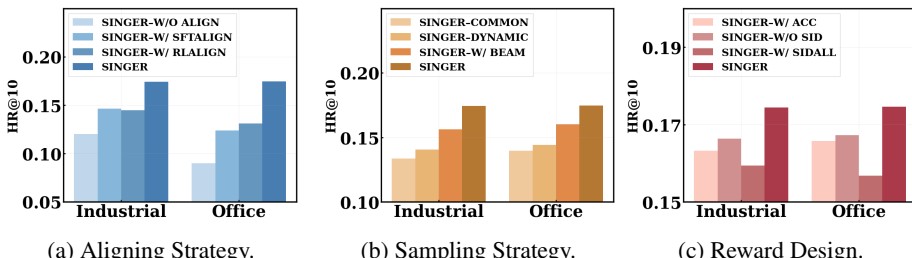

(a) Aligning Strategy.    (b) Sampling Strategy.    (c) Reward Design.

Figure 4: Study on the effectiveness of SINGER's individual components. Figure 4a examines model performance under different alignment strategies; Figure 4b investigates various sampling strategies; Figure 4c evaluates the impact of alternative reward designs.

Alignment tasks are used only during the SFT stage, whereas the RL stage is trained on SID-only data. (3) SINGER–W/ RLALIGN: The SFT stage relies on SID-only supervision, while alignment tasks are introduced solely in the RL stage.

As illustrated in Figure 4a, the full SINGER with full-process SID alignment achieves the best results across all metrics. The SINGER–W/O ALIGN variant performs the worst, underscoring the importance of grounding world knowledge when generating SIDs. Notably, although introducing the alignment objective directly in the RL stage is highly challenging for an LLM that has not been pre-conditioned by SFT, SINGER–W/ RLALIGN still yields a non-trivial gain. We attribute this improvement to our SID-Guided RL scheme, which offers the agent additional opportunities to obtain valid rewards on hard examples that conventional RL would likely miss.

### 4.3.2 Sampling Strategy

We contrast the full model with three variants that differ only in the way trajectories are collected: (1) SINGER–COMMON that uses a conventional top-$k$ decoding scheme to generate the required number of trajectories. (2) SINGER–DYNAMIC, which implements our dynamic sampler that first produces $\frac{3}{2}$ times the target trajectory budget and then keeps as many distinct items as possible for RL optimization. (3) SINGER–W/ BEAM that retains only beam search; no SID-prefix guidance is applied, and all examples maintain their original difficulty level.

As Figure 4b illustrates, the full SINGER achieves the best overall performance. Moreover, the SINGER–W/ BEAM variant attains higher accuracy than SINGER–DYNAMIC while requiring only two-thirds of its sampling budget, showing that beam search is a more cost-effective backbone. These findings motivate our final design that merges beam search with SID-prefix guidance.

### 4.3.3 Reward Design

Three variants are compared: (1) SINGER–W/ ACC that uses the accuracy reward only; (2) SINGER–W/O SID that removes the SID-level reward for hard cases; the LLM is optimized with the accuracy plus rank rewards. (3) SINGER–W/ SIDALL that applies the SID-level reward to every training sample rather than restricting it to difficult ones.

As shown in the Figure 4c, the full model, which deploys SID-level reward only for hard samples, achieves the best overall performance. To be noticed, rewarding all samples at the SID-level slightly degrades performance. We hypothesize that, for easy instances already covered by the accuracy and ranking rewards, an additional SID-level signal may dilute the optimization focus and introduce a mismatch with the evaluation metrics (HR@K and NDCG@K). For hard cases, however, the hierarchical structure encoded in SIDs provides intermediate guidance where conventional rewards are sparse, steering the model toward correct reasoning paths.

## 5 CONCLUSION

This paper investigates how the *SFT-then-RL* paradigm, which has recently proved successful in language–reasoning tasks, can be adapted to the generative recommendation setting. Through a careful analysis, we identify two central obstacles — limited SID understanding and ineffective

reward assignment — that prevent a direct transfer of the vanilla pipeline. To overcome these issues, we propose **SINGER**, a SID-Navigated GEnerative Recommender that (i) performs Full-Process SID Alignment to inject SID-aware objectives into every stage of post-training, and (ii) introduces SID-Navigated RL, which supplies fine-grained SID-level rewards and a hierarchy-based curriculum sampler. Experiments on two public benchmarks demonstrate consistent improvements over SOTA sequential, generative, and LLM-based recommenders, showing that deep SID comprehension and SID-Navigated RL feedback are both indispensable for unleashing the full potential of LLMs in recommendation.

CLAIM

## 5.1 ETHICS STATEMENT

Our study relies exclusively on publicly available benchmark datasets that contain no personally-identifiable information. Data collection, storage, and processing strictly follow the licences and terms of use provided by the original publishers. All LLMs employed in this work are open-sourced and used under their respective permissive licences. We make no attempt to infer sensitive attributes of users, and all generated recommendations are produced within a controlled, offline research environment. The authors declare that no conflict of interest exists.

## 5.2 REPRODUCIBILITY STATEMENT

We take reproducibility seriously and adopt the following measures: 1) All source code, configuration files, and experiment scripts will be released upon publication. 2) We provide detailed instructions for environment setup, including package versions, CUDA/driver requirements. 3) Random seeds are fixed for data splitting, parameter initialization, and sampling operations. 4) Pre-processed datasets, together with the raw-to-processed conversion scripts, are included to guarantee identical data partitions.

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

# A  RELATED WORK

## A.1  GENERATIVE RECOMMENDATION

In recent years, generative recommendation has attracted considerable attention in both academia and industry. This emerging paradigm, usually built on the Transformer architecture, reformulates recommendation as an end-to-end next-item generation task and thereby raises the performance ceiling of recommender systems. Early work TIGER (Rajput et al., 2023) employs residual quantization (RQ-VAE) (Zeghidour et al., 2022) to convert text embeddings—extracted from an item's title and description—into discrete semantic IDs, which are then used for next-item prediction. HSTU (Zhai et al., 2024) propose a new architecture designed for high cardinality, non-stationary streaming recommendation data. LC-Rec (Zheng et al., 2024) aligns an LLM with semantic IDs via multi-task learning, enabling the model to "understand" the IDs and perform generative recommendation. Other studies investigate how to build better semantic IDs to enhance generation quality: RecForest (Feng et al., 2022) applies hierarchical k-means clustering and treats the cluster indices as tokens, while EAGER (Wang et al., 2024) and TokenRec (Qu et al., 2024) integrate semantic and collaborative signals directly into the tokenizer.

Very recently, generative recommendation has been rolled out at an industrial scale to address the drawbacks of traditional cascade systems. MTGR (Wang et al., 2025a) keeps the original deep learning recommendation model (DLRM) features, introduces user-level compression, and speeds up both training and inference for large-scale deployment. OneRec (Deng et al., 2025) lowers the serving cost with a Lazy Decoder-Only Architecture and stabilises training through an improved reinforcement learning algorithm.

## A.2  LLM AND RL

Reinforcement learning (RL) trains an agent through repeated interaction with an environment so as to maximise cumulative return (Kaelbling et al., 1996; Sutton and Barto, 2018). Within large-language-model (LLM) fine-tuning, RL with Human Feedback (RLHF) has become the de-facto recipe: it usually adopts Proximal Policy Optimisation (PPO) (Schulman et al., 2017) to align model behaviours with human preferences (Kaufmann et al., 2023). Unfortunately, PPO is memory-hungry at the billion-parameter scale, motivating a series of lighter alternatives. Direct Preference Optimisation (DPO) (Rafailov et al., 2023) removes the value network and directly maximises the log-likelihood gap between preferred and dispreferred outputs; s-DPO (Chen et al., 2024) adapts this idea to recommendation by casting softmax negative mining as a pairwise-preference signal. Yet, preference-based methods remain off-policy and often plateau below on-line RL. Group-Relative Policy Optimisation (GRPO) (Shao et al., 2024) mitigates memory cost by normalising rewards inside a small group of roll-outs and replaces a learned reward model with rule-based heuristics, achieving strong gains on reasoning-heavy tasks such as mathematics and programming (DeepSeek-AI et al., 2025; OpenAI et al., 2024).

Recent studies have begun to explore how SFT and RL jointly shape LLMs for generative recommendation. (Yoshihara et al., 2025) argues that the two stages are complementary: a prolonged SFT phase first pushes accuracy to its limit, after which on-line RL with GRPO further compresses the token budget at inference time. (Jin et al., 2025) shows that RL can largely recover the out-of-distribution accuracy lost during SFT by cancelling the directional drift of singular vectors rather than by discovering entirely new solutions. In contrast, (Yue et al., 2025) points out that the reasoning ability obtained through RL with verifiable rewards (RLVR) is bounded by the base model, whereas SFT can introduce genuinely new reasoning patterns, suggesting the need for more powerful RL paradigms such as continual scaling. (Cheng et al., 2025) adds a domain perspective: areas frequently encountered during pre-training (e.g., mathematics and code) profit from cross-domain RL, while low-exposure domains (e.g., logic and simulation) require in-domain RL for meaningful gains. (Zhao et al., 2025) observes that popular RL algorithms tend to converge to a single dominant output distribution, amplifying patterns already present in the pre-training data, yet they still display cross-task generalization.

```
Input:
The user has interacted with items <a_13><b_197><c_1>, <a_52><b_17><c_113>, <a_13><b_201><c_34> in
chronological order. Can you predict the next possible item that the user may expect?

Response:
<a_13><b_72><c_149>
```

Figure 5: Semantic task prompt.

```
Input:
What is the title of <a_24><b_141><c_73>?

Response:
Oral-B Deep Sweep Toothbrush
```

Figure 6: Alignment task prompt1.

## B EXPERIMENTAL SETTINGS

All conventional recommender baselines are optimized with binary cross-entropy (BCE) loss and the Adam optimizer. The learning rate is selected from $\{1\times10^{-2}, 1\times10^{-3}, 1\times10^{-4}\}$, while the weight-decay coefficient is tuned within $\{1\times10^{-2}, 1\times10^{-3}, 1\times10^{-4}, 1\times10^{-5}, 1\times10^{-6}\}$. A mini-batch size of 1024 is used throughout. For TIGER, we adopt T5 (Sanh et al., 2022) as the encoder–decoder backbone and use Qwen3-Embedding-4B to generate item embedding. Every LLM-based method, including ours, is built upon Qwen2.5-Instruct-0.5B (Yang et al., 2024) to keep the computational footprint modest, and is trained with the AdamW optimizer.

The SFT and preference-alignment data are processed in batches of 128, whereas RL batches contain 512 samples. We set the learning rate to $3\times10^{-4}$ for SFT and to $1\times10^{-5}$ for both S-DPO and SINGER, together with a cosine decay scheduler. SFT runs for ten epochs with early stopping (patience = 1). S-DPO is trained for a single epoch, and we fix $\beta = 0.1$ and sample three negative items. For $D^3$, the interpolation coefficient $\alpha$ is chosen from $\{0.8, 0.9, 1.0\}$.

For the SID generation stage of SINGER, we utilize Qwen3-Embedding-4B as the text encoder to transform item titles and descriptions into their corresponding embeddings. The tokenizer is trained on 8 GPUs with a per-device batch size of 2048. RQ-VAE is trained layer-wise for 1 000 steps per layer, with a learning rate of $1\times10^{-3}$. We employ a Constrained Balanced RQ-KMeans algorithm to generate SIDs. Specifically, we perform residual quantization layer-wise with a codebook size of $K = 256$. To prevent cluster collapse and maximize codebook utilization, we enforce strict size constraints on each cluster, ensuring a balanced tree structure. The clustering is optimized for a maximum of 100 iterations per layer with a convergence tolerance of $1 \times 10^{-7}$. Crucially, to ensure a strictly one-to-one mapping between items and SIDs, we apply a deterministic deduplication step: for any items sharing the same semantic path, a unique suffix token is appended to resolve conflicts. Following SID generation, the SFT stage is conducted with a batch size of 128 for up to ten epochs (early stopping, patience = 1), followed by the full-process alignment-guided RL for two epoch under the same $\beta$ and candidate settings as described above.

## C DATASETS

We evaluate our approach on two subsets of the Amazon Review corpus: *Industrial_and_Scientific* and *Office_Products*. To keep the computational cost manageable, we adopt a data-reduction procedure

```
Input:
Which item has the title Nashua Stretch & Seal Self-Fusing Silicone Tape?

Response:
<a_202><b_202><c_29>
```

Figure 7: Alignment task prompt2.

Input:
The user has interacted with items 'Kreg SML-C150-100 Pocket Screws 1-1/2-Inch, 8 Coarse, Washer-Head, 100-Count', '3M Flap Disc 566A, T29, 4-1/2"" Diameter, 40 Grit, 5/8""-11 Thread (Pack of 1)' in chronological order. Can you predict the next possible item that the user may expect?

Response:
<a_104><b_60><c_152>

Figure 8: Alignment task prompt3.

Input:
The user has interacted with items '<a_71><b_44><c_249>', '<a_71><b_114><c_136>', '<a_67><b_244><c_35>' in chronological order. Can you predict the title of the next item that the user may expect?

Response:
Install Bay Copper Ring Terminal Connector 8 Gauge 5/16 Inch 25 Pack - CUR8516']",J-B Weld 8265S Original Cold-Weld Steel Reinforced Epoxy - 2 oz.

Figure 9: Alignment task prompt4.

inspired by the strategy in (Bao et al., 2024). The preprocessing steps are as follows: (1) users and items with fewer than five interactions are removed; (2) for the *Toys_and_Games* subset, only records from October 2016 to November 2018 are retained; (3) for the smaller *Industrial_and_Scientific* subset, we keep all interactions between October 1996 and November 2018; (4) each user's interaction sequence is truncated to a maximum length of ten; (5) finally, each dataset is split chronologically into training, validation, and test partitions with an 8:1:1 ratio. The main statistics of the resulting training splits are listed in Table 3.

Table 3: Statistics of datasets.

| Datasets | Inductrial | Office |
|----------|-----------|--------|
| Items | 3,685 | 3,459 |
| Train | 3,6259 | 3,8924 |
| Valid | 4,532 | 4,866 |
| Test | 4,533 | 4,866 |

## D  ALIGNMENT PROMPTS

For the semantic task, a sample prompt is shown in the Figure 5.

Representative examples for the alignment tasks are presented in Figures 7, 6, 8, 9, and 10.

## E  LIMITATION

Although SINGER delivers promising results, our study is subject to several limitations. First, limited computational resources prevent us from fully probing the gains that might be unlocked when scaling the framework to larger language models and larger datasets; consequently, the upper bound of

Input:
An item can be described as follows: Tach-It B-1 single-edge industrial razor blades (pack of 100) are high-carbon steel replacement blades designed for heavy-duty cutting and scraping tasks; they fit most standard single-edge blade holders and come packaged in a convenient bulk dispenser. Which item is it describing?

Response:
<a_17><b_91><c_139>

Figure 10: Alignment task prompt5.

SINGER's performance remains unclear. Second, despite its potential, we did not investigate how to systematically improve the cross-domain performance of the SINGER-w/o-SFT variant. These limitations indicate that future work should explore the scalability and robustness of SINGER in more resource-intensive and iterative settings, thereby offering a more comprehensive assessment of its practical usefulness.

## F  LLM USAGE

LLMs were employed exclusively for linguistic polishing and stylistic refinement of the manuscript. No LLM was used to generate experimental designs, implement algorithms, produce empirical results, or analyze data. All technical contributions, including model architecture, training protocol, and evaluation pipeline, were conceived, implemented, and verified by the authors themselves.

