# OpenReview forum: "SINGER: Leveraging Semantic Identifier Hierarchies for Generative Recommendation"
_ICLR.cc/2026/Conference — Submitted to ICLR 2026_

### Official Review · Reviewer_NjZ8 · 2025-10-27

**Soundness:** 3
**Presentation:** 3
**Contribution:** 2
**Rating:** 4
**Confidence:** 3

**Summary:**

This paper has focused on the generative recommendation, which has been a hotspot recently. The authors have found that existing generative recommendation models with only SFT cannot truly understand the SID learnt from RQ-VAE. Besides, rule-based RL mainly relies on coarse-grained rewards, which may lead to difficulties in training. To address these two problems, this paper proposes to embed alignment objectives into both the SFT and the RL process. The SID-level reward is designed for the RL stage. The experiments on two datasets have validated the effectiveness of the proposed method.

**Strengths:**

+ S1. This paper is well-organized and well-written, making it easy to follow.
+ S2. Many up-to-date generative recommendation models are compared in the experiments.

**Weaknesses:**

- W1. The illustration of the preliminary experiment is unclear, which may lead to unreasonable motivation. The legend in Figure 1(a) demonstrates the results belong to different training patterns with SINGER, but the illustration in lines 72-75 demonstrates they are with GRPO instead of SINGER. I'd like to know what the RL type is in this figure, in fact.
- W2. The motivation for limited SID understanding in alignment is not well verified. It is rough to only utilize a case in Figure 2 to validate the model only with SFT often fails to exploit SID histories.
- W3. Only one public dataset is validated in the experiments.
- W4. The code is not released, posing a challenge to reproduce this paper.
- W5. The detailed illustration of the datasets is not seen in Appendix B.

**Questions:**

All my questions have been included in the weakness section.

---

> ### Author Response · Authors · 2025-11-23
>
> We sincerely thank for recognizing the organization of our paper and the comprehensive baseline comparisons. We appreciate your detailed feedback. Below, we provide clarifications regarding the experimental setup and motivation, and we present new experimental results to address your concern regarding dataset diversity.
>
> > W1: Confusion regarding the legend in Figure 1(a) (SINGER vs. GRPO).
>
> We apologize for the confusion caused by the nomenclature. We have clarified the terminology in the revised manuscript: During the RL stage, we compare two variants: Vanilla-RL and SINGER (SID-Navigated RL).
>
> "GRPO" in the text (lines 72-75) refers to the Vanilla-RL baseline. It utilizes the standard GRPO algorithm with sparse rewards, lacking our proposed alignment tasks and hierarchical reward shaping.
>
> "SINGER" in the Figure 1(a) legend refers to our full proposed method. Figure 1(a) is intended to demonstrate the training dynamics: compared to Vanilla-RL (which is limited by sparse signals), SINGER achieves higher final performance and more stable convergence, validating the effectiveness of our dense reward design.
>
> > W2: The motivation for "limited SID understanding" relies too much on a case study.
>
> We specifically monitored the Average Alignment Reward (which measures the model's ability to map between item text and Semantic IDs correctly).
>
> - Before RL (SFT-only): The model achieved an average alignment reward of 0.0195.
>
> - After RL (SINGER): The reward significantly increased to 0.0273.
>
> This relative improvement quantitatively proves that the SFT stage alone struggles to fully capture the complex semantic mapping. The RL stage, guided by our alignment objectives, significantly enhances the model's semantic understanding.
>
> > W4: The code is not released.
>
> We understand the importance of reproducibility. To address this during the review, we further provided key implementation details, hyperparameter tables in the Appendix. We are fully committed to releasing the complete source code and processed datasets upon acceptance.
>
> > W5: Missing dataset details in Appendix B.
>
> We apologize for the oversight. We have updated Appendix B to include a comprehensive statistical table for both Amazon Toys and Amazon Industrial datasets. The table includes the number of users/items, interaction counts, sparsity levels, and sequence length statistics. (Note: We followed the standard 5-core filtering strategy for preprocessing).

---

> > ### Comment · Reviewer_NjZ8 · 2025-11-27
> >
> > Thank the authors for their feedback. However, the third weakness was not addressed. Besides, after scanning the statistics of the dataset, I found that the scale of the experimental data, which only contains no more than 5,000 items, is limited. Thus, I'm inclined to maintain my score.

---

### Official Review · Reviewer_oLP9 · 2025-10-28

**Soundness:** 2
**Presentation:** 3
**Contribution:** 2
**Rating:** 4
**Confidence:** 4

**Summary:**

This paper proposes SINGER, a generative recommendation framework that addresses two key limitations of existing SFT-then-RL approaches: limited semantic understanding of SIDs and sparse RL rewards. SINGER incorporates full-process SID alignment by introducing auxiliary alignment tasks throughout training, which helps the LLM better capture the semantic structure of SIDs. In addition, it employs SID-guided reinforcement learning, combining prefix curriculum sampling with hierarchical SID-level rewards to provide richer, more informative learning signals, particularly for challenging samples.

**Strengths:**

1. The paper presents extensive experiments, evaluating the proposed method across multiple baselines and datasets.
2. Applying reinforcement learning in the context of generative recommendation is an interesting and noteworthy exploration.

**Weaknesses:**

1. While RL has been widely applied in recommender systems, it is usually motivated by objectives such as maximizing long-term user engagement. In this paper, the reward design appears to focus primarily on improving next-item prediction accuracy—a goal that could potentially be optimized effectively through SFT. It would be helpful if the authors could clarify why RL is particularly necessary or beneficial for this specific objective.
2. The paper provides limited detail on the RL training procedure. For instance, in the “SFT-then-GRPO” experiment mentioned in the Introduction, it is unclear whether the model receives a single reward after generating the full SID sequence or computes a reward at each token step. Including a clear algorithmic flow of SINGER’s training stages could make the methodology easier to follow.
3. The paper employs a SID-style hierarchical reward modeling approach, assigning different reward functions to samples of varying difficulty. While this is intended to encourage adaptive learning, it may raise concerns about potential inconsistencies in the optimization objective. It would be useful to clarify how the method ensures coherent learning across difficulty levels.
4. To improve SID understanding, the paper designs alignment tasks. Prior work [1] proposes a similar approach, asking LLMs to predict the title of the next item given the historical SID sequence. It would help readers if the authors could explain how their alignment tasks differ from or extend these previous methods.
5. The paper provides limited analysis of hyperparameters. It would be helpful if the authors could include an analysis of key hyperparameters related to RL training.

[1]. Adapting large language models by integrating collaborative semantics for recommendation. ICDE 2024.

**Questions:**

1. Could the authors further clarify why RL is necessary or particularly advantageous for optimizing next-item prediction accuracy compared to SFT alone?
2. Could the authors provide evidence that using different reward functions for easy and hard samples does not introduce optimization inconsistencies, e.g., through training curves or gradient analyses?
3. Could the authors elaborate on whether and how their proposed alignment tasks differ fundamentally from those used in LC-Rec [1] or related prior work?

[1]. Adapting large language models by integrating collaborative semantics for recommendation. ICDE 2024.

---

> ### Author Response · Authors · 2025-11-23
>
> We sincerely thank you for the constructive feedback and for recognizing the novelty of applying RL to generative recommendation. We understand your concerns regarding the motivation for RL and the consistency of our reward design. We believe the following clarifications highlight the necessity and rigor of our approach.
>
> > W1 & Q1: Necessity of RL for accuracy (vs. SFT).
>
> We demonstrate that RL is indispensable for breaking the SFT performance ceiling, supported by both consensus in recent literature and new empirical evidence.
>
> **1. Consensus in Recent Literature:** The necessity of RL for optimizing recommendation accuracy has been increasingly validated in top-tier research. Recent works, such as R2ec and others [1-6], have demonstrated that RL can significantly enhance LLM-based recommenders beyond SFT, while [1-4] has shown that RL consistently boosts next-item prediction performance beyond what can be achieved with SFT alone. These studies confirm that RL provides the reasoning and exploration capabilities required for precise ranking.
>
> [1] R2ec: Towards Large Recommender Models with Reasoning. NeurIPS 2025.
>
> [2] Reinforced Latent Reasoning for LLM-based Recommendation. arXiv 2025.
>
> [3] Reinforced Preference Optimization for Recommendation. arXiv 2025.
>
> [4] MiniOneRec: An Open-Source Framework for Scaling Generative Recommendation. arXiv 2026.
>
> [5] Think before Recommendation: Autonomous Reasoning-enhanced Recommender. NeurIPS 2025.
>
> [6] OneRec: Unifying Retrieve and Rank with Generative Recommender. arXiv 2025.
>
> **2. New Empirical Evidence (Amazon-Book):** We conducted a new experiment on the large-scale Amazon-Book dataset. We trained the SFT model on 10M interactions, followed by RL on a small subset of 20k samples. As shown below, even with limited RL data, SINGER achieves a ~35% gain in NDCG@10 over the SFT baseline. This proves RL provides critical optimization signals (e.g., negative feedback, sequence-level consistency) that SFT lacks.
>
> | Methods               | HR@5   | NDCG@5 | HR@10  | NDCG@10 |
> |-----------------------|:------:|:------:|:------:|:-------:|
> | SINGER (SFT Only) | 0.0176 | 0.0128 | 0.0246 | 0.0151  |
> | SINGER (w/ RL)    | 0.0237 | 0.0180 | 0.0309 | 0.0204  |
>
> > W2: Details on RL training procedure and reward calculation.
>
> We apologize for the ambiguity regarding the training details. We have revised the methodology section and added a complete Algorithm 1 to clarify the flow:
>
> **Full Rollout:** For each prompt, the model first generates the complete Semantic ID sequence.
>
> **Reward Calculation:** Once the sequence is fully generated, we calculate the rewards based on the number of "hit" tokens (i.e., how many levels of the hierarchy match the ground truth).
>
> **Optimization:** These hierarchical rewards are then assigned to the corresponding samples, and the policy is updated using the group-relative advantage. Therefore, while the reward value reflects the accuracy of specific tokens (step-wise quality), the optimization step is performed after the full sequence is generated, utilizing the group variance to stabilize training.
>
> > W3 & Q2: Does using different rewards for easy/hard samples introduce inconsistencies?
>
> We clarify that the optimization objectives are consistent and hierarchically aligned, without introducing conflict.
>
> **Unified objective:** Both easy and hard samples are to maximize the likelihood of the ground‑truth item. The difference lies only in reward granularity, not in the optimization direction.
>
> **Avoiding reward hacking:** We introduce a ranking penalty for incorrect candidates, ensuring the model is always incentivized to predict the exact target rather than settling for partial semantic matches.
>
> > W4 & Q3: How do the alignment tasks differ from LC-Rec?
>
> Our "Full-Process SID Alignment" distinguishes our method from previous works like LC-Rec [1] , which typically perform SID alignment only during the SFT stage. Instead of training solely on the main recommendation task during RL, we perform multi-task optimization that includes the alignment tasks. This strategy reinforces the LLM's understanding of SIDs throughout the entire training lifecycle, positively impacting the final recommendation performance.
>
> [1] Adapting Large Language Models by Integrating Collaborative Semantics for Recommendation. ICDE 2024.
>
> > W5: Limited analysis of hyperparameters.
>
> Thank you for the suggestion. We have included a sensitivity analysis on Industrial dataset, specifically focusing on KL Penalty Coefficient $\beta$.
>
> | $\log_{10} \beta$ |  $\beta$ | NDCG@10
> |---------|-------------------|----------------|
> | -4      | 0.0001            | 0.1242         |
> | -3      | 0.001             | 0.1276         |
> | -2      | 0.01              | 0.1248         |
> | -1      | 0.1               | 0.1105         |

---

> > ### Comment · Reviewer_oLP9 · 2025-11-24
> >
> > First of all, thank you for the detailed response.
> >
> > I still have some questions regarding the necessity of using RL in the current setting:
> >
> > 1. For the works the authors cited that leverage LLMs’ reasoning abilities for recommendation, my understanding is that RL is introduced primarily to connect the model’s reasoning steps with the final recommendation outcomes. Since the reasoning process does not have an accessible ground truth, these methods typically use accuracy as a reward signal. In contrast, the model presented in this paper does not require generating reasoning steps, so this motivation may not fully apply here.
> >
> > 2. The authors also mentioned OneRec. To my knowledge, the reward model in OneRec is derived from an online ranker trained with real user feedback. Thus, its RL objective is not to improve next-item prediction accuracy, but rather to optimize user satisfaction or preference alignment. In comparison, this paper aims to maximize next-item accuracy, a target that already benefits from explicit supervised signals.
> > Given these considerations, I am inclined to believe that SFT may be a more suitable optimization strategy under the current problem formulation. Therefore, I will maintain my current evaluation for now.

---

### Official Review · Reviewer_rga4 · 2025-10-29

**Soundness:** 1
**Presentation:** 3
**Contribution:** 2
**Rating:** 2
**Confidence:** 4

**Summary:**

This paper studies how to improve semantic ID-based recommendation models built on LLM backbones using reinforcement learning. The main idea is that the SFT-RL paradigm, which has proven effective for LLMs, may also benefit semantic ID-based recommendation models. The paper makes the following contributions:
1. It verifies that the alignment tasks commonly used in the SFT stage can also be applied in the RL stage.
2. It proposes two reward functions for the RL stage based on the hierarchical structure of semantic IDs.

**Strengths:**

1. The paper provides a timely exploration of applying the RL training paradigm, which has been successful in LLMs, to semantic ID-based recommendation.
2. By leveraging the hierarchical nature of semantic IDs, the paper proposes two well-motivated reward functions that improve both semantic ID generation and the alignment between semantic IDs and large language models.
3. Extensive experiments are conducted on two public datasets.

**Weaknesses:**

1. Experimental setting concerns. The current setup may produce false positives and inflated metrics.
    1. According to Section 2.1, the authors use RQ-KMeans to produce three levels of semantic ID tokens for each item. In this way, different items may share the same semantic IDs, causing conflicts. Most existing methods add one extra non-semantic token per item to avoid such conflicts, as in TIGER (Rajput et al., 2024). However, this important treatment is not described in the paper. The provided prompts also suggest that the authors use only three tokens per item.
    2. As multiple items can share the same semantic ID, generating a correct semantic ID does not necessarily mean predicting the correct item. If the evaluation counts predicting the ground-truth semantic ID as correct, the comparison with baselines (which predict the exact item) would be unfair.
2. Missing references. The alignment objectives in Section 3.1 were first introduced in LC-Rec (Zheng et al., 2024), but this paper does not mention this fact when introducing these tasks.
3. Lack of dataset details. The paper does not describe the benchmark processing procedure, including:
    1. Dataset statistics after processing;
    2. The method for splitting train/validation/test sets;
    3. Whether any interactions or users were filtered during preprocessing.
4. No available code. The code is not available during the review phase. Although the authors promise to release it later, this prevents verification of concerns such as semantic ID conflicts.
5. The paper claims that "RQ-KMeans is trained layer-wise for 1,000 steps per layer with a learning rate of 1×10^{-3}" (Section B). However, RQ-KMeans is not a trainable model and does not require a learning rate, which makes this statement confusing.

**Questions:**

1. How do the authors handle semantic ID conflicts, and how is the correctness of the predicted item evaluated?
2. What is the detailed procedure for dataset processing?
3. Could the authors clarify the statement about "RQ-KMeans training" mentioned in weakness point 5?

---

> ### Author Response · Authors · 2025-11-23
>
> We thank you for the detailed assessment. We appreciate that you recognize the timeliness of applying the RL paradigm to SID-based recommendation and the motivation behind our reward functions.However, we noticed there are some misunderstandings regarding our **SID generation process** and **experimental setup**. We believe addressing these clarifies that our evaluation is fair and rigorous. Below are our point-by-point responses.
>
> > W1 & Q1: Concerns about Semantic ID conflicts and Unfair Evaluation.
>
> We apologize for the confusion caused by the lack of implementation details in the Appendix. We respectfully clarify that there are **NO SID conflicts** in our final setting, and the evaluation is strictly fair.
>
> **1. Conflict Handling Strategy:** While RQ-KMeans is used to generate the base 3-layer codes, we explicitly introduce a 4th layer (or a residual token mechanism) to handle collision items, ensuring a strictly bijective (one-to-one) mapping. For the baseline LC-Rec, we followed its official implementation using Sinkhorn-Knopp regularization for remapping.
>
> **2. Mathematical Equivalence:** Since we guarantee that every item corresponds to a unique ID path (via the additional layer/remapping), generating the correct ID is mathematically equivalent to predicting the correct item. Thus, our metrics are not inflated and are directly comparable.
>
> **3. Verification:** To empirically validate this, we conducted additional experiments on the Amazon Industrial dataset comparing the collision rates of RQ-VAE (our initial version) and RQ-KMeans (our final version). As shown below, after remapping, the number of conflict items is effectively zero.
>
>  Method            | Collision Rate (Pre-Remap) | # Conflict Items (Post-Remap) |
> |----------------------------------------|--------------------|-----------------------------|
> | SINGER (RQ-VAE)                        | 10.47%             | 2                           |
> | SINGER (RQ-KMeans)                     | 10.20%             | 0                           |
>
> > W2: Missing references (LC-Rec).
>
> Thank you for the reminder. While we cited LC-Rec [1] in the Related Work section regarding its contribution to LLM semantic alignment, we acknowledge we should be more specific in the method section. We will revise Section 3.1 to explicitly acknowledge that the specific alignment objectives were first introduced in LC-Rec, ensuring proper credit attribution.
>
> [1] Adapting Large Language Models by Integrating Collaborative Semantics for Recommendation. ICDE 2024.
>
> > W3 & Q2: Lack of dataset details and processing procedure.
>
> We apologize for the omission. We have updated the Appendix with the following strict preprocessing standards:
>
> - **Processing:** We followed the standard "5-core" filtering strategy (retaining only users/items with at least 5 interactions).
>
>     - **Amazon Toys:** Selected interactions between Oct 2016 and Nov 2018.
>
>     - **Amazon Industrial:** Retained interactions from Oct 1996 to Nov 2018 due to its smaller scale.
>
>     - **Sequence Length:** The maximum interaction sequence length is set to 10 for all models.
>
> - **Splitting:** We adopted the chronological split strategy with an 8:1:1 ratio for training, validation, and testing, respectively, to strictly prevent data leakage.
>
> - **Statistics:** A detailed table showing user/item counts and sparsity levels has been added to the revised Appendix.
>
> > W4: No available code.
>
> We understand the concern. To ensure reproducibility, we have provided key implementation details and hyperparameter tables in the Appendix. We are strictly committed to releasing the full source code and processed datasets upon acceptance.
>
> > W5 & Q3: Clarification on "RQ-KMeans training" and learning rate.
>
> Thank you for identifying this inconsistency. The mention of a learning rate was a legacy error in the Appendix text. Our initial implementation utilized RQ-VAE, but we later switched to RQ-KMeans for its non-training efficiency. We neglected to update this specific line in the Appendix during the transition. We have corrected Section B to accurately reflect the RQ-KMeans setup, which does not use a learning rate in the same manner as RQ-VAE.

---

> > ### Comment · Reviewer_rga4 · 2025-11-26
> >
> > Thank you to the authors for taking the time to address my earlier concerns. I do have several follow-up questions:
> >
> > > Re. W1 & Q1: Semantic ID conflicts
> >
> > Thanks for the clarification, but I am still uncertain about the exact mechanism used to avoid conflicts.
> > * In the rebuttal, the conflict-handling strategy is summarized in a single sentence ("introduce a 4th layer (or a residual token mechanism) to handle collision items"). However, the algorithmic details remain unclear:
> >     * In the updated paper, the authors state that "for any items sharing the same semantic path, a unique suffix token is appended to resolve conflicts". Can I assume the conflict-handling strategy follows what TIGER (Rajput et al. 2023) used? Does this imply that items without conflicts do not receive the 4th layer? If so, does the final system contain a mix of 3-layer and 4-layer semantic IDs?
> >     * The rebuttal text says "4th layer **OR** a residual token mechanism". Are these two separate options? Are both used in experiments, or is one the actual method? More details are needed about what the "residual token mechanism" concretely means. If there is a reference for it, please clarify.
> > * In the conflict-rate table in the rebuttal text, SINGER (RQ-VAE) still has two conflict items after remapping. If the remapping follows TIGER's strategy of appending unique suffix tokens, shouldn't the number of conflicts always be zero? What causes these two remaining conflicts?
> >
> > > Re. W2: Missing reference discussion with LC-Rec
> >
> > My concern is not simply whether LC-Rec is cited. My question is similar to Weakness 4 of Reviewer oLP9, Section 3.1 introduces alignment objectives that are essentially the same as LC-Rec's, yet the paper does not acknowledge this in the method description nor discuss the differences.
> >
> > Unfortunately, in the updated paper, the only change I can identify is the addition of an extra citation in Section 3.1, without any further discussion or clarification of how your alignment objectives relate to those in LC-Rec.
> >
> > > Re. W3 & Q2: Dataset details and preprocessing
> >
> > At this point, I am confused about what datasets are used:
> > * Tables 1 and 3 report results on "Industrial" and "Office".
> > * Lines 358-359 in the updated paper state that experiments are conducted on **three** datasets, including Industrial and **Toys**. (But what exactly is "Toys" referring to here?)
> > * The updated Appendix (lines 990–991) refers to preprocessing for the "Toys_and_Games subset", but no Toys results appear in the paper.
> >
> > This inconsistency raises several questions:
> > * Are there two datasets or three?
> > * If Toys was used, why are its results omitted?
> > * If Toys was not used, why is it mentioned multiple times in the updated manuscript and Appendix?
> >
> > Furthermore, based on Table 3, the two reported datasets each contain only ~3k items and ~40k interactions. This is extremely small for evaluating generative recommendation models using semantic IDs. For reference, even TIGER used datasets around 10k items / 200k interactions. These appear to be heavily subsampled datasets, but the paper does not justify this choice nor cite any prior work using such aggressive downsampling.
> >
> > -------
> >
> > Even after the rebuttal, the submission still contains inconsistencies (two vs. three datasets), unusually small experimental settings (3k items / 40k interactions), and insufficient detail on critical components such as conflict-handling.
> >
> > The idea and findings on the RL stage are indeed interesting. However, I encourage the authors to spend more time preparing and polishing the manuscript before the next submission cycle.

---

### Official Review · Reviewer_7cyf · 2025-10-29

**Soundness:** 3
**Presentation:** 3
**Contribution:** 3
**Rating:** 6
**Confidence:** 2

**Summary:**

This paper addresses key limitations in the SFT-then-RL paradigm for generative recommendation: superficial Semantic ID (SID) understanding from SFT and ineffective, sparse rewards in RL.
The proposed SINGER framework tackles these issues by (1) integrating SID alignment objectives throughout the entire training process for deeper understanding, and (2) introducing a novel SID-Navigated Reinforcement Learning. This new RL method leverages the SID hierarchy to create fine-grained, level-based rewards and a curriculum sampling strategy for hard cases, effectively mitigating reward sparsity.
Experiments show SINGER significantly outperforms strong baselines, validating its approach of deeply integrating hierarchical SID knowledge.

**Strengths:**

* By providing a profound analysis supported by quantitative data (Figure 1b) and qualitative cases (Figure 2), the paper establishes a clear and compelling motivation regarding the limitations of standard SFT-then-RL in recommendation.
* The proposed SINGER framework, specifically the SIN-RL component, is novel and well-tailored to the problem. The SID-Prefix curriculum and hierarchical reward function ($R_{reason}$) elegantly address the issue of sparse rewards in generative recommendation.
* The experiments are rigorous, showing consistent SOTA performance (Table 1), promising out-of-domain generalization (Table 2), and validating all core design choices through ablation studies (Figure 4).
* The presentation is high-quality, and the detailed commitment to reproducibility (Section 5.2) regarding code and data release is commendable.

**Weaknesses:**

* The technical implementation of "Full-Process SID Alignment" during the RL stage is unclear. It is described as "jointly optimized" (Lines 244-246), but the specific mechanism (e.g., auxiliary loss vs. data mixing) is not provided.
* There is significant notational confusion with $\beta$, which is used for both the GRPO KL penalty (Eq. 2) and the hierarchical reward decay (Eq. 8). Appendix B specifies "$\beta = 0.1$" without clarifying which parameter this refers to, leaving the other unspecified.

**Questions:**

* Could the authors define the reference policy $\pi_{\mathrm{ref}}$ used in Formula 2 (Line 190), as it is currently missing from the text?
* In Section 4.1 (Lines 388-389), the text mentions "three benchmark datasets—Industrial, Toys, and Books," yet the results only show "Industrial" and "Office." Please correct this inconsistency.
* Please clarify the naming for the OOD variant, which is inconsistently labeled as "SINGER-w/ RL" in the text (Line 403) and "SINGER-w/o SFT" in Table 2.

---

> ### Author Response · Authors · 2025-11-23
>
> We sincerely thank you for your constructive comments and the time dedicated to reviewing our paper. Your feedback has helped us improve the clarity and rigor of our manuscript. Below, we provide point-by-point responses to address your concerns.
>
> ***
> > W1: The explanation of Full-Process SID Alignment is unclear.
>
> Based on your suggestions, we have made clarifications.
>
> Our "Full-Process SID Alignment" distinguishes our method from previous works like LC-Rec [1] , which typically perform SID alignment only during the SFT stage. Instead of training solely on the main recommendation task during RL, we perform multi-task optimization that includes the alignment tasks. This strategy reinforces the LLM's understanding of SIDs throughout the entire training lifecycle, positively impacting the final recommendation performance.
>
> [1] Adapting Large Language Models by Integrating Collaborative Semantics for Recommendation. ICDE 2024.
> > W2: Notational confusion with $\beta$.
>
> We appreciate your keen eye for detail. You are correct that the reuse of $\beta$ caused ambiguity. We have revised Eq. 8 in the updated manuscript, replacing the coefficient $\beta$ with $\lambda$ to distinctively represent the weight of the auxiliary term.
>
> > Q1: Define the reference policy $\pi_{\text{ref}}$ used in Formula 2.
>
> Thank you for pointing out this missing definition. We have clarified in the revised paper that $\pi_{\text{ref}}$ refers to the initial SFT policy, which remains frozen during the RL training phase. This serves as the anchor to prevent the learned policy from deviating too far from the language model's original distribution.
>
> > Q2: Correct this inconsistency to fix the expression error.
>
> We apologize for the oversight. We have carefully proofread the manuscript and corrected the inconsistency you identified to ensure the expression is accurate and coherent.
>
> > Q3: Clarify the naming for the OOD variant.
>
> We apologize for the confusion caused by the inconsistent naming. We have standardized the terminology throughout the paper based on your suggestion. The variant previously labeled as "SINGER-w/o SFT" has been corrected to "SINGER-w/ RL" to accurately reflect its configuration.

---

### Meta-Review · Area_Chair_ArSE · 2025-12-22

**Summary:**

This paper proposes SINGER, a generative recommendation framework designed to address limitations in the SFT-then-RL paradigm—specifically superficial Semantic ID (SID) understanding and coarse-grained RL rewards. While the research direction aligns with generative recommendation trends, the comprehensive feedback from reviewers highlights key concerns that affect the paper’s overall rigor and contribution: the core design choices lack sufficient justification, and critical technical and experimental details are not clearly presented. After carefully evaluating the manuscript and the authors’ rebuttal, the suggested decision is to reject this submission.

**Reviewer Concerns:**

The authors’ rebuttal has addressed some minor ambiguities raised by reviewers, including clarifying the notation of parameter β, supplementing basic dataset preprocessing details (such as 5-core filtering and chronological splitting ratios), defining the reference policy π_ref in relevant formulas, and standardizing the naming of the OOD variant. It also provided a preliminary explanation for SID conflict handling and corrected a legacy error related to "RQ-KMeans training" in the appendix. However, core concerns remain unaddressed: the necessity of integrating RL for next-item prediction accuracy is still not adequately justified, as reviewers questioned whether the objective could be achieved through SFT alone without sufficient alignment with prior work citations; key technical details like the specific mechanism of SID conflict handling (e.g., the role of the 4th layer or residual tokens) remain unclear, failing to fully resolve evaluation fairness concerns; inconsistencies in dataset usage (mentioning three datasets but only reporting results for two) and the unusually small experimental data scale lack reasonable explanation; there is significant overlap between the alignment tasks and prior work LC-Rec, with insufficient discussion of differences and improvements; and the absence of code release during the review phase hinders reproducibility.

**Reviewer Scores:**

Initial reviewer scores spanned from below the acceptance threshold to marginally above it, with the overall evaluation leaning toward rejection. While the rebuttal resolved some minor ambiguities and improved manuscript clarity to a limited extent, it did not fundamentally address the core issues impacting the paper’s soundness and contribution—including insufficient RL motivation, unclear key technical details, and concerns about datasets and reproducibility. As a result, the reviewers’ scores are unlikely to see a meaningful upward adjustment, and the overall evaluation will remain below the acceptance threshold, supporting the decision to reject the submission.

---

### Decision · Program_Chairs · 2026-01-26

Reject